# Recycling of Uridylated mRNAs in Starfish Embryos

**DOI:** 10.3390/biom14121610

**Published:** 2024-12-16

**Authors:** Haruka Yamazaki, Megumi Furuichi, Mikoto Katagiri, Rei Kajitani, Takehiko Itoh, Kazuyoshi Chiba

**Affiliations:** 1Department of Biological Sciences, Ochanomizu University, Bunkyo-ku, Tokyo 112-8610, Japanshin1126t628@gmail.com (M.K.); 2School of Life Science and Technology, Institute of Science Tokyo, Meguro-ku, Tokyo 152-8550, Japan; rei.kajitani@gmail.com (R.K.); takehiko@life.isct.ac.jp (T.I.)

**Keywords:** poly(A), mRNA degradation, deadenylation, uridylation, ribosomal protein mRNAs, starfish

## Abstract

In eukaryotes, mRNAs with long poly(A) tails are translationally active, but deadenylation and uridylation of these tails generally cause mRNA degradation. However, the fate of uridylated mRNAs that are not degraded quickly remains obscure. Here, using tail-seq and microinjection of the 3′ region of mRNA, we report that some mRNAs in starfish are re-polyadenylated to be translationally active after deadenylation and uridylation. In oocytes, uridylated maternal cyclin B mRNAs are stable without decay, and they are polyadenylated to be translated after hormonal stimulation to resume meiosis, whereas they are deadenylated and re-uridylated at the blastula stage, followed by decay. Similarly, deadenylated and uridylated maternal ribosomal protein mRNAs, *Rps29* and *Rpl27a*, were stable and inactive after hormonal stimulation, but they had been polyadenylated and active before hormonal stimulation. At the morula stage, uridylated maternal ribosomal protein mRNAs were re-polyadenylated, rendering them translationally active. These results indicate that uridylated mRNAs in starfish exist in a poised state, allowing them to be recycled or decayed.

## 1. Introduction

In eukaryotes, most mRNAs carry a non-templated poly(A) tail at the 3′ end, and modifications to this tail are important for the regulation of mRNA stability, transport, and translation. For example, long poly(A) tails can interact with poly(A) binding protein (PABP), which protects mRNAs from degradation, and they can also initiate translation by interacting with the 5′ cap-bound translation initiation factor 4F complex (eIF4F) [1]. The shortening of the poly(A) tail is conducted by deadenylases, PAN2-PAN3 and/or the CCR4–NOT complex, which have overlapping functions [2]. Subsequently, uridine residues are added to 3′ ends of short poly(A) tails by terminal uridylyl transferase 4 (TUT4) and TUT7 [3,4,5]. mRNAs having oligo(U) tails are degraded by the 5′–3′ exonuclease XRN1 [3,6] or the 3′–5′ exonuclease Dis3L2 [3,7,8]. Deletion of TUT4/7 genes in mice or depletion of TUT4/7 in zebrafish and *Xenopus* leads to developmental defects and blocks mRNA decay [4,9]. In yeast *Schizosaccharomyces pombe*, a Dis3L2 homolog was found to degrade uridylated RNA substrates [10], indicating that deadenylation and uridylation of mRNA tails constitutes a conserved system of RNA degradation in eukaryotes [11,12].

To accumulate many ribosomes required for early embryonic development, ribosomal protein mRNAs with long poly(A) tails are translated within growing animal oocytes arrested at prophase of meiosis I (Pro-I) [13]. Following hormonal stimulation to resume meiosis, ribosomal protein mRNAs are deadenylated to become translationally inactive, as observed in *Xenopus* oocytes [14]. However, deadenylation of mRNAs upon hormonal stimulation does not induce decapping or degradation during meiotic division; rather, deadenylated mRNAs are stable until the blastula stage [14,15,16,17]. These results suggest that additional steps may exist between deadenylation (or possibly uridylation) and decay of these mRNAs, and the mechanism by which stability is conferred on transcripts with short poly(A) tails following deadenylation still needs to be determined [18]. In addition, mRNAs with short poly(A) tails from different genes have widely different decay rate constants (1000-fold) in mammalian culture cells [19], indicating that mRNAs with short poly(A) tails are not always rapidly decayed. Moreover, in mammalian somatic cells, nearly half of the mRNA species are uridylated at a >5% frequency, and SOGA2 and PABPC4 mRNAs are more highly uridylated at frequencies of 41% and 24%, respectively [20]. These results suggest that uridylated mRNA species may have different decay rate constants.

Poly(A) tails of maternal mRNAs in animal oocytes, such as cyclin B, are short but are stably stored. In *Xenopus* oocytes, poly(A) lengths of *cyclin B1* mRNAs are controlled by cytoplasmic polyadenylation elements (CPEs) in their 3′ UTRs and CPE binding protein (CPEB), recruiting poly(A) polymerase (Gld2), and deadenylating enzyme (PARN) [21,22,23,24]. Poly(A) tails are kept short by PARN and Gld2 in dynamic equilibrium until hormonal stimulation [25]. Upon meiosis resumption, CPEB phosphorylation occurs, which induces PARN release, causing poly(A) elongation and translational activation. Previously, we found that the uridylation of cyclin B mRNA at the 3′ end of a short poly(A) tail does not cause degradation of the mRNA in starfish oocytes arrested at Pro-I [26]. After hormonal stimulation with 1-methyladenine (1-MA), which activates the Cdk1–cyclin B complex through the starfish serum- and glucocorticoid-regulated kinase (SGK)-dependent pathway [27,28], some uridine residues of *cyclin B* mRNA were trimmed and the 3′ ends were subsequently polyadenylated [26]. These results suggest that uridylation has a function other than the degradation of mRNA in starfish oocytes and that the mechanism of shortening poly(A) tail lengths differs from that in *Xenopus* oocytes.

In this study, we investigated the fate of maternal *cyclin B* and ribosomal protein mRNAs during starfish development. As a result of the maternal-to-zygotic transition (MZT), new mRNA synthesis does not occur until the blastula stage in starfish [29,30]. This long physiological block of transcription uniquely enables the tracing of sole maternal mRNA modifications throughout development. However, in vertebrates, MZT occurs as early as the two-cell stage [31], complicating the distinction between modifications of maternal transcripts and newly synthesized zygotic mRNA. Using the starfish system, we revealed that uridylated ribosomal protein mRNAs are re-polyadenylated and translationally reactivated at the morula–blastula stage, while uridylated *cyclin B* mRNAs decay after MZT. These results provide new insights into mechanisms underlying the recycling of uridylated mRNA.

## 2. Materials and Methods

### 2.1. Animal and Oocyte Preparation

Starfish (*Asterina pectinifera*) were collected from the Pacific coast of Japan during the breeding season and were maintained in laboratory aquaria with seawater at 14 °C. This study was conducted in accordance with ARRIVE guidelines for animal research. To remove follicle cells, oocytes, released from isolated ovaries, were washed three times with ice-cold Ca^2+^-free seawater (450 mM NaCl, 9 mM KCl, 48 mM MgSO_4_, 6 mM NaHCO_3_, 40 mM 3-[4-(2-hydroxyethyl)-1-piperazinyl] propanesulfonic acid (EPPS), pH 8.0) and incubated in artificial seawater (450 mM NaCl, 9 mM KCl, 48 mM MgSO_4_, 6 mM NaHCO_3_, 40 mM EPPS, 9.2 mM CaCl_2_, pH 8.0) or Jamarin artificial sea water (JAMARIN-U, Jamarin Laboratory, Osaka, Japan) at 20 °C. Meiosis resumption of starfish oocytes was induced using 1 µM 1-methyladenine (1-MA). After verifying the breakdown of the germinal vesicle, “dry” sperm obtained from male starfish was added at a final dilution of 1/100,000, after which 1-MA and sperm were washed out by changing seawater. All experiments were performed at 20 °C unless otherwise stated. Total RNA from oocytes was extracted using the RNeasy mini kit (QIAGEN, Hilden, The Netherlands) and quantified using a Qubit 3.0 fluorometer (Thermo Fisher Scientific, Waltham, MA, USA) as per manufacturer’s instructions.

### 2.2. RT-PCR with 3′ Adaptor Ligation and Tail Sequence

Synthesis of cDNA with a biotinylated 3′ adaptor ligation was performed using a small RNA cloning kit (Takara Bio Inc., Kusatsu, Japan), as previously reported [26,32]. In some experiments, the thermostable group II intron reverse transcriptase (TGIRT) template-switching RNA-seq kit (InGex, Olivette, MO, USA) was used for reverse transcription as per manufacturer’s guidelines (Appendix A). PCR was performed using gene-specific primers and a 3′ adaptor primer (primer sets are listed in Appendix A). PCR products were purified by agarose gel electrophoresis and extracted from the gel using the Wizard SV Gel and PCR Clean-Up System (Promega, Madison, WI, USA). Purified PCR products were cloned into a pCR2.1-TOPO vector (Invitrogen, Carlsbad, CA, USA), and insert sequences were determined using Sanger sequencing with the Applied Biosystems™ 3130 DNA Analyzers (Applied Biosystems, Waltham, MA, USA).

### 2.3. 5′-Rapid Amplification of cDNA

The *Rps29* 5′ UTR and the coding region cDNAs were synthesized using the SMART™ RACE cDNA Amplification Kit (Clontech Laboratories Inc., Mountain View, CA, USA) as per the manufacturer’s guidelines. PCR was performed using gene-specific primers (40S 5′ RACE R, Appendix A) and a 5′ primer (SMART oligo) provided by the manufacturer. PCR products were purified and cloned into a pCR2.1-TOPO vector as described in Section 2.2.

### 2.4. DNA Cloning and Plasmids for In Vitro RNA Synthesis

*psfcycB_A25*: To synthesize an RNA encompassing 3′ UTR of starfish *cyclin B* carrying a unique sequence tag and a long poly(A) tail without uridines, we modified the pcDNA used in our previous study [26]. A 25 nt poly(A) sequence was inserted after the 3′ UTR using the In-Fusion (Takara Bio) method.

*psfRps29_WT*: To create the pcDNA including sfRps29 3′ UTR, 3′ UTR of starfish *Rps29* cDNA were amplified by PCR using an sfRps29_F2 primer and PCR-R&RT primer, and the product was TA-cloned into vector pCR2.1-TOPO (Invitrogen, Carlsbad, CA, USA) as described in Section 2.2, followed by another PCR. The starfish *Rps29* 3′ UTR cDNA was amplified to produce fragment 1 using primers, psfRps29_WT_vector_F containing an exogenous sequence tag, and psfRps29_WT_vector_R. Fragment 2, containing the 3′ region of the 3′ UTR of starfish *Rps29* cDNA, 20 bases of poly(A), and a BsmBI restriction endonuclease site, was produced using PCR with the psfRps29_WT_insert_template, and primers psfRps29_WT_insert_F and psfRps29_WT_insert_R. Fragment 3 was pBluescript KS (-) (Stratagene, CA, USA), which was digested using Xba. These three fragments were ligated using In-Fusion (Takara Bio Inc., Shiga, Japan), and the resultant plasmid was named psfRps29_WT.

*psfRps29*_*PAS*: The Delta PAS sfRps29 vector was generated by PCR using PrimeSTAR MAX Polymerase (Takara Bio) with primers containing point mutations (forward: 5′-TCAGAAAGAAAATGACCAGATCTGCT-3′, reverse: 5′-TCATTTTCTTTCTGAACTCAATACAC-3′), and the template psfRps29_WT. Point mutation sites in the primers are underlined.

*psfSGK vector*: The psfSGK vector included T7 promoter, *A. pectinifera cyclin B* Kozak sequence (59-TACAAT-39), sfSGK (T479E), and a C-terminal 3 × FLAG tag in the pSP64-S-based vector [28,33,34]. To alter the 3′ terminal sequence and insert the BbsI site, we generated double-stranded DNA from single-stranded oligo DNAs via the 5′–3′ polymerase activity of PCR polymerase and inserted the product into the psfSGK vector using the In-Fusion method. The primers and oligo DNAs are listed in Appendix A.

*psf Rps29 UTR luciferase vector*: The *Rps29* 5′ UTR and the coding region cDNAs in the pCR2.1-TOPO vector were amplified via PCR using primers sfRps29-GST_1F and sfRps29-GST_1R. The GST was amplified via PCR using primers sfRps29-GST_2F and sfRps29-GST_2R. The psfRps29 WT vector, including the 3′ UTR of *Rps29* mRNA, was amplified via PCR using primers sfRps29-GST_3F and sfRps29-GST_3R. They were then ligated using In-Fusion (Takara Bio Inc., Shiga, Japan), and the resultant plasmid was named psfRps29-GST. Then, the GST region of psfRps29-GST was replaced with luciferase cDNA (pNL1.1[Nluc]Vector; Promega Madison, WI, USA). Luciferase cDNA was amplified via PCR using primers Nluc_FW2 and Nluc_RV2. The psfRps29-GST was amplified via PCR using primers vec_FW2 and vec_RV2. These PCR fragments were ligated using In-Fusion, and the resultant plasmid was named psf Rps29 UTR Luciferase, which was used to produce the reporter mRNA.

### 2.5. Microinjection

Microinjection was performed as previously described [26,35]. Briefly, the oocytes were injected using a constricting pipet filled with in vitro synthesized RNA (10–100 ng/µL) (injection volume, 1–2% oocyte volume). Injected oocytes were incubated for the indicated periods; 100 oocytes were used for each experiment. Injection timing and workflow are shown (Appendix A).

### 2.6. In Vitro Transcription

The in vitro transcription of starfish *cyclin B* (*sfcyclin B*) mRNA was performed as previously described [26] using the mMESSAGE mMACHINE T7 kit (Invitrogen) with the original NTP concentrations as per the manufacturer’s guidelines. Transcripts were subsequently polyadenylated using the Poly(A) Tailing Kit (Thermo Fisher Scientific) as per the manufacturer’s guidelines and purified using phenol/chloroform extraction and ethanol precipitation.

The DNA templates for in vitro transcription of sfRps29_WT mRNA and sfRps29_delta PAS mRNA were PCR-amplified from the pRps29_WT vector or pRps29_PAS vector, respectively, using a universal M13 forward primer (5′-GTAAAACGACGGCCAGT-3′) and universal M13 reverse primer (5′-CAGGAAACAGCTATGAC-3′) (Takara Bio Inc., Shiga, Japan). The PCR product was digested with BsmBI (Esp3I), and the desired fragment was purified using agarose gel electrophoresis with the Wizard SV Gel and the PCR Clean-Up System followed by phenol/chloroform extraction and ethanol precipitation, with the resultant DNA pellet subsequently dissolved in sterile water. Using this DNA as a template, in vitro transcription was performed using the mMESSAGE mMACHINE T7 kit with the original NTP concentrations as per the manufacturer’s guidelines. Transcripts were subsequently polyadenylated using the Poly(A) Tailing Kit as per the manufacturer’s guidelines and purified by phenol/chloroform extraction and ethanol precipitation.

The DNA templates for SGK were amplified using psfSGK_RNA_F (5′-GCCAGATGCTACACAATTAGGC-3′) and psfSGK_RNA_R (5′-TTGTTTGCCGGATCAAGAGC-3′) primers. PCR products were digested with BbsI (BpiI) and polyadenylation using a Poly(A) Tailing Kit as per the manufacturer’s guidelines when long poly(A) was required.

The psf Rps29 UTR luciferase vector was digested with BsmBI and purified via agarose gel electrophoresis using the Wizard SV Gel followed by phenol/chloroform extraction, ethanol precipitation, in vitro transcription, and poly(A) elongation as described above.

### 2.7. Real-Time qPCR

Total RNA was extracted using the RNeasy Micro Kit (QIAGEN), and 300 ng was reverse-transcribed using an iScript Select cDNA synthesis kit (Bio-Rad, Berkeley, CA, USA) and random hexamers as per the manufacturer’s guidelines. The cDNAs were quantified with SYBR-Green assays (Applied Biosystems, Foster City, CA, USA) using the Applied Biosystems 7500 Real-Time PCR System; 18S rRNA was used for normalization. The primers used for qRT-PCR are listed in Appendix A.

### 2.8. Translation Assay

SGK mRNAs were injected into the oocytes. Oocytes incubated in Jamarin seawater for the indicated periods were recovered in 3 µL of seawater, added directly to 3 µL of 2 × Laemmli sample buffer, and immediately frozen in liquid N_2_. After thawing and boiling for 5 min at 95 °C, the proteins were separated on 8% polyacrylamide gels and immunoblotted (primary antibody: anti-sfSGK-HM (1:1000, in TBS-T) [27,28]; secondary antibody: HRP-conjugated anti-rabbit IgG (A6154, Sigma; 1:2000, in TBS-T)). Proteins recognized by the antibodies were visualized using ECL prime (GE Healthcare, Chicago, IL, USA), and digital images were acquired with a LAS4000 mini-imager (FUJIFILM Wako Pure Chemical, Osaka, Japan).

### 2.9. Luciferase Assay

Eggs and embryos were lysed using the Reporter Lysis Buffer (Promega, Madison, WI, USA). Luminescence was assessed using the Nano-Glo^®^ Luciferase Assay System (Promega, Madison, WI, USA) and quantified using the Lumicounter 700 (MICROTEC Co., Ltd., Funabashi, Japan).

### 2.10. Illumina MiSeq Sequencing for Targeted TAIL-Seq

The multiplexed cDNA samples were sequenced using Illumina MiSeq (San Diego, CA, USA) (Appendix A). Sequencing was performed to obtain paired-end reads (2 × 300 nt) according to the manufacturer’s protocol. The quality filter was turned off to avoid removing reads comprising poly(A), which often includes low-quality bases. To prevent saturation from similar sequences, DNA samples from other organisms were mixed with the starfish-derived cDNAs to decrease the ratio of the cDNA samples to 22%. It is of note that reads from these organisms have distinct sequences from the target cDNA of the starfish and, therefore, could be separated using different barcodes (indices) and excluded from downstream analysis.

### 2.11. Correction of mRNA (cDNA) Sequences for Mapping of Targeted TAIL-Seq Reads

To increase the mapped reads and prevent the misdetection of 3′ tail modifications, the mRNA (cDNA) sequences used as references for read mapping were modified (Appendix A). The target genes were *Rps29* (40S ribosomal protein) and *Rpl27a* (60S ribosomal protein). The procedure for each sample was as follows:(1)To prevent the misalignments of reads, a poly(A) sequence 50 nt in length was added to the 3′ end of the mRNA in silico.(2)Reads of the GV sample from the oocytes before hormonal stimulation were randomly downsampled to a coverage depth of 1000×.(3)Downsampled reads were mapped to the mRNA sequence using BWA-MEM [36] (version 0.7.12-r1044) with the default parameters.(4)Mapped reads were piled up using SAMtools [36] (version 1.3.1). The commands were ‘sort’, ‘index’, and ‘mpileup’ with the parameters, and the output was the text file of the pileup result.(5)For each position in the mRNA sequence, a base (A, T (U), G, or C) was changed to the majority position using the pileup result and the in-house program.(6)The poly(A) sequence added to (1) was removed.

### 2.12. Extraction of Valid 3′ Reads from the Targeted TAIL-Seq

For each sample, the procedures were as follows (the processes without tool names were performed using in-house programs) (Appendix A):(1)The 3′ adapters in the reads were searched for using BLASTN (Camacho et al., 2009 [37]) (version 2.6.0) with the option of ‘-task blastn-short -word_size 6 -window_size 0 -dust no’ to handle short sequences. Here, the query and database inputs were reads and adapters, respectively (‘-query reads.fa -db adapter’).(2)Reads in which 3′ adapters were aligned with identity ≥ 90%, in which alignment-coverage of the adapter was 100%, and in which alignment positions were the heads of reads were extracted. Read pairs without the 3′ adapter were discarded. For each extracted read pair, the reads with and without the 3′ adapter were treated as 3′ and 5′ reads, respectively.(3)The 5′ reads were aligned to the targeted mRNA using BLASTN with the options of ‘task blastn -word_size 10 -window_size 0 -dust no’ and read pairs in which 5′ reads were aligned with identity ≥ 90% and alignment length ≥ 200 nt were extracted, and the others were discarded to exclude contamination.

### 2.13. Analysis of Poly(A) and 3′ Modifications in Targeted TAIL-Seq Reads

The procedures were based on the methods introduced in the initial TAIL-Seq study [20] with some modifications (Appendix A). Reads were treated as FASTQ files, without applying the previous methods utilizing raw signal information, to analyze non-canonical poly(A) sequences. Poly (A) was sequenced as poly(T) as the 3′ reads corresponded to the reverse complements of the mRNA sequences. The following procedures targeted the reverse complements of the 3′ reads to provide concise descriptions and were performed using the in-house programs:(1)The 3′ reads of the extracted pairs (see ‘Extraction of 3′ reads of targeted mRNAs’) were aligned to the targeted mRNA using BLASTN with the options of ‘-task blastn -word_size 10 -window_size 0 -dust no’.(2)If an alignment with a length ≥ 30 nt was detected in a 3′ read, the aligned region was treated as a 3′ UTR (not a 3′ tail region). To handle low-quality reads associated with poly(A), a sequence identity cut-off was not applied.(3)To detect poly(A) regions, a score was calculated based on the composition of bases as follows: A, +1; N, −1; T, G, or C, −2. If the score was higher, it was considered that the region was more likely to comprise poly(T). Although the concept of this score was introduced in a previous study [20], our calculation method differed from that originally published (A, +1; N, −2; T, G, or C, −10) to detect non-canonical poly(A) regions. For each 3′ read, the scores were calculated for all regions (substrings), and the region that satisfied the following conditions was determined as a poly(A): maximum score throughout the read, score > 0, and distance to the 3′ end of the read ≤ 15 nt.(4)Using the results of (2) and (3), regions in each 3′ read were classified as 3′ UTR, poly(A), 3′ end modification (a region between poly(A) and the 3′ end of a read), and other. Statistics, such as length, were calculated for each class. When calculating the length distributions of poly(A)s, the regions with lengths ≥ 40 nt were treated as the same category (‘≥40 nt’) because the lengths of the long poly(A)s tended to be overestimated due to systematic base-calling errors [20].(5)For each 3′-end modification, classification was based on the major base (T, G, or C). If multiple bases occurred at the same time, the modification was classified as ‘≥2′.(6)To reduce the influence of sequencing errors, bases in the 3′ reads with quality values of <20 were masked, and the composition of the bases (rates of A, T, G, and C) was calculated using unmasked reads.

### 2.14. Analysis of TAIL-Seq Data of Xenopus Laevis

The TAIL-Seq data for *X. laevis* were generated in a previous study [4]. The targeted sample set comprised the wild-type *X. laevis* early embryos (internal ID, hs27), which comprised embryos from the zygote (1 cell) to stage 12. The procedures were as follows (processes without tool names were performed using in-house programs):(1)The data set was generated and pre-processed in a previous study [4] using Tailseeker (version 3.1.7; https://github.com/hyeshik/tailseeker, accessed on 13 September 2019); the pre-processing steps included the removal of adaptors and PCR duplicates. Intermediate FASTQ files of paired reads were obtained through personal communication with Dr. Hyeshik Chang (an author of the previous study).(2)The 5′ reads were mapped to the *X. laevis* RNA sequence set (RefSeq accession, GCF_001663975.1; file name, GCF_001663975.1_Xenopus_laevis_v2_rna_from_genomic.fna) using BWA-MEM (Li and Durbin, 2009 [36]) (version 0.7.12-r1044) with the default parameters.(3)The 5′ reads that were primarily mapped to the mRNA of *rps29* (40S ribosomal protein; accession, NM_001171730.1) were detected and the corresponding 3′ reads extracted.(4)For the extracted 3′ reads, the same procedures were applied as for the starfish (see ‘Detection and analysis of 3′ ends of mRNA including poly(A)’).

In addition, we also attempted to analyze the TAIL-Seq data from *X. laevis* oocytes (internal ID, ms97); however, the number of reads that mapped to the *rps29* mRNA was too low to analyze.

## 3. Results

### 3.1. Cylin B mRNA Decay Occurs After Deadenylation and Uridylation at MZT

Deadenylation and uridylation of maternal mRNAs, including *cyclin B2* in *Xenopus* and zebrafish embryos, induce mRNA decay at MZT [4,38]. In starfish, the expression of many zygotic genes, including *wnt8* and *foxq2*, which are marker genes for MZT, is initiated at the blastula stage [29,30,39]. To determine whether maternal *cyclin B* mRNAs in starfish are degraded during meiosis resumption, fertilization, or MZT, we used real-time qPCR (RT-qPCR) to monitor their maternal expression levels before fertilization in Pro-I and MI oocytes forming the first polar body (1PB), or after fertilization in embryos at morula, blastula, and gastrula stages. We found that maternal *cyclin B* mRNA levels were decreased in gastrula embryos (Figure 1A), suggesting that the decay of *cyclin B* mRNAs occurred at MZT between the blastula and gastrula stages. In addition, these results indicated that *cyclin B* mRNAs in Pro-I oocytes do not decrease following meiosis resumption, although they carry uridylated short poly(A) tails [26].

To monitor tail lengths and sequences of the endogenous maternal *cyclin B* mRNAs during meiosis resumption, fertilization, and development, cDNAs containing an adapter ligated to 3′ ends were produced using a template-switching reaction catalyzed by TGIRT-III enzyme (Appendix A). We then performed RT-PCR using a 3′ adaptor primer and a *cyclin B*-specific primer designed to hybridize 373 nt upstream of the polyadenylation site (sfcycB F primer; Appendix A).

Polyacrylamide gel electrophoresis of RT-PCR products, including a 3′ adapter sequence, yielded a single 420 bp band from oocytes at the Pro-I stage of meiosis (Figure 1C, left panel, endogenous). This band was stably detected in unstimulated oocytes incubated in seawater for 24 h after isolation. In comparison, after hormonal stimulation, 450–550 bp mobility-shifted bands were produced (Figure 1C, left panel: 1 and 6 h, endogenous). At the blastula stage, after 12 h of hormonal stimulation, shorter bands appeared (Figure 1C left panel, 12 h, endogenous), but then disappeared in gastrula stage embryos (Figure 1C, left panel, 24 h, endogenous).

As previously reported [26], Sanger sequencing of RT-PCR products showed that endogenous *cyclin B* mRNAs in oocytes before hormonal stimulation carried uridylated short poly(A) tails (Figure 1D, endogenous −1-MA) and were polyadenylated after hormonal stimulation (Figure 1D, endogenous +1-MA 1 h).

These long poly(A) tails were stable until the morula stage (Figure 1C, left panel, 6 h, endogenousAppendix A, Endogenous, +1-MA 6 h Morula). Furthermore, at the blastula stage, short poly(A) tails were uridylated (Figure 1D, endogenous +1-MA 12 h), followed by the decay of *cyclin B* mRNAs at the MZT between the blastula and gastrula stages (Figure 1A).

As some uridine residues originating from oligo(U) tails in Pro-I oocytes were included in long polyadenylated tails of *cyclin B* mRNA following hormonal stimulation [26] (see also Figure 1D, endogenous +1-MA 1 h 1PB), they could be exposed at 3′ ends of the tails through trimming or partial deadenylation of long poly(A) tail at the 12 h blastula stage. To distinguish these ‘old’ uridines exposed upon deadenylation at the blastula stage from possible ‘new’ uridylation during development, we synthesized an exogenous RNA encoding the 3′ UTR of starfish *cyclin B* carrying both a unique sequence tag and a long poly(A) tail without uridines (Figure 1B, upper panel). When this RNA was injected into oocytes following hormonal stimulation, lengths of the RT-PCR products from this exogenous mRNA became shorter at the blastula stage and disappeared at the gastrula stage (Figure 1C, right panel), showing a pattern comparable to that of the endogenous *cyclin B* mRNA (Figure 1C, left panel). In addition, Sanger sequencing revealed that exogenous mRNAs were deadenylated and uridylated at 12 h in the blastula stage (Figure 1E), followed by the decay of mRNA (Figure 1C, right panel). These results suggest that uridylation of *cyclin B* mRNA at the blastula stage in starfish embryos promotes its degradation, as seen in vertebrate embryos [4]. They also show that the 3′ UTR of *cyclin B* alone is enough to drive its deadenylation and uridylation.

### 3.2. Ribosomal Protein mRNAs Are Uridylated After 1-MA Stimulation and Fertilization, Followed by Non-Canonical Poly(A) Tail Formation in Embryos at the Blastula Stage

As ribosomes are required for the translation of maternal mRNAs, such as that of *cyclin B*, mRNAs of ribosomal proteins carrying long poly(A) tails are actively translated in oocytes to produce ribosomes before hormonal stimulation. Following hormonal stimulation to resume meiosis, deadenylation of ribosomal protein mRNAs leads to cessation of ribosome production, as shown in *Xenopus* oocytes [13,14], although mRNA of the ribosomal proteins L5 and L13 could still be detected until stage 6 of early blastula [40,41]. These results suggest that the deadenylation of ribosomal protein mRNAs does not always lead to mRNA decay. Indeed, expression levels of starfish ribosomal protein mRNA (*Rps29*), evaluated using RT-qPCR, did not decrease in stimulated oocytes or the embryos at morula and blastula stages (Figure 2A). Instead, the mRNA level increased at the blastula stage, which was likely due to the zygotic expression of *Rps29* mRNA, as was previously shown in *Xenopus* embryos [41].

To determine whether 3′ ends of ribosomal protein mRNAs of starfish oocytes or embryos were modified during the resumption of meiosis and early development, total RNAs were purified at indicated times (Figure 2B), and then a 3′ adaptor was ligated to their 3′ ends [26] (Appendix A). When 3′ ends of the 40S ribosomal protein *Rps29* mRNA were amplified using a gene-specific primer and a 3′ adaptor primer, polyacrylamide gel electrophoresis of RT-PCR products showed that the broad bands (320–470 bp) apparent in the oocytes before hormonal stimulation (Figure 2B, 0 h after 1-MA stimulation) had become shorter and sharper (310 bp) following hormonal stimulation (Figure 2B, 2 h after 1-MA stimulation). This shortening of PCR product lengths may be due to deadenylation induced by the 1-MA stimulation, as decreases in poly(A) lengths of ribosomal proteins are well reported in *Xenopus* oocytes after meiotic resumption [14]. Consistent with this, Sanger sequencing showed that more than half of the *Rps29* mRNAs before hormonal stimulation contained long poly(A) tails (Figure 2C, −1-MA), whereas all mRNAs in stimulated oocytes carried short poly(A) or uridylated short poly(A) tails (Figure 2C, +1-MA, 1.5 h). Furthermore, even before hormonal stimulation, some *Rps29* mRNAs carried short uridylated poly(A) tails (Figure 2C, −1-MA), which likely became shorter after hormonal stimulation (Figure 2C, +1-MA, 1.5 h).

Notably, the lengths of the RT-PCR products became longer at the morula and blastula stages (Figure 2B), whereas the amount of mRNA did not increase at these stages (Figure 2A). Sanger sequencing of PCR products revealed that long poly(A) tails of the mRNAs from the blastula stage included many U and some G nucleotides (Figure 2C, +1-MA, 12 h). We defined such tails as non-canonical poly(A) tails. Similarly, mRNA of the 60S ribosomal protein L27a (*Rpl27a*) was deadenylated and uridylated after hormonal stimulation (Appendix A, −1-MA; +1-MA, 1.5 h). In the blastula stage, the *Rpl27a* mRNAs carried non-canonical poly(A) tails, which included many GU nucleotides (Appendix A, +1-MA, 12 h).

### 3.3. Targeted TAIL-Seq of Rps29 mRNA

To obtain more detailed quantitative and qualitative information about tail structures of the *Rps29* mRNA, we modified the ‘TAIL-Seq’ method by Lim et al. [3] to perform targeted RNA sequencing, which was achieved using an amplicon-based approach with a gene-specific primer for *Rps29* mRNA and the 3′ adaptor primer. We could detect two peaks representing long poly(A) tails (>40 nt) and short poly(A) tails (10–20 nt) in oocytes before hormonal stimulation (Figure 3A, −1-MA), as observed using Sanger sequencing (Figure 2C, −1-MA). Long poly(A) tails in the unstimulated oocytes were not uridylated (Figure 3B, −1-MA, >40), whereas short poly(A) tails (10–20 nt) were highly uridylated (>60%) (Figure 3B, −1-MA, ≤40). Following hormonal stimulation, these two populations decreased significantly and a new peak (0–10 nt) appeared (Figure 3A, +1-MA, 1.5 h), the tails of which were uridylated (Figure 2C and Figure 3B, +1-MA, ≤40). These results suggest that 1-MA stimulation led to the shortening of both long poly(A) tails and short uridylated poly(A) tails, followed by new uridylation of short poly(A) tails during meiosis resumption.

The frequency of non-A residues appeared to be higher in the 5′ region of non-canonical poly (A) tails of *Rps29* mRNA in the blastula stage (Figure 2C, +1-MA, 12 h). Consequently, percentages of non-A residues were calculated at indicated positions in tails from 5′ to 3′ using the results of the TAIL-Seq (Figure 3C, left panel). As expected, approximately 30% of residues were non-A in the 5′ region, with percentages decreasing gradually as the number of residues increased toward the 3′ direction. Furthermore, in tails from 3′ to 5′ for embryos at the blastula stage (Figure 3C, right panel), the frequencies of non-A residues were lower than those from 5′ to 3′ (Figure 3C, left panel), indicating that 3′ end region of non-canonical poly(A) tails contain more A residues than the 5′ end region. Frequencies of uridine or guanine were higher than those of cytosine in non-canonical poly(A) tails (Figure 3D). These results suggest that non-canonical poly(A) tail construction was initiated by the formation of a U-rich tail followed by an A-rich tail elongation. When percentages of non-A residues were calculated in unstimulated oocytes from 3′ to 5′ (Figure 3C, right panel), frequency at the 3′ end in oocytes before hormonal stimulation was approximately 40%, which may be due to uridylated short poly(A)s (10–20 nt) carrying additional U tails (Figure 2C, endogenous −1-MA; Figure 3B, −1-MA, ≤40). Similar results were obtained when a targeted TAIL-Seq was performed using the specific primer for *Rpl27a* mRNA (Appendix A).

### 3.4. Injected 3′ UTR of Rps29 mRNA Behaves Similarly to Endogenous mRNA

To confirm whether 40S *Rps29* mRNA containing non-canonical poly(A) tails originated from maternal mRNA containing long canonical poly(A), but not from zygotic mRNA transcribed during embryogenesis, 3′ UTR of *Rps29* mRNA was synthesized having both a unique sequence tag and a long poly(A) tail and injected into oocytes (Figure 4A). If maternal *Rps29* mRNA was deadenylated after 1-MA stimulation and re-adenylated in embryos at morula and/or blastula stages, exogenous 3′ UTRs of *Rps29* mRNA carrying long poly(A) tails would be deadenylated after hormonal stimulation, followed by re-elongation of the poly(A) tails. As expected, electrophoresis of RT-PCR products of the endogenous and exogenous *Rps29* mRNA tails showed that their lengths decreased similarly at 1 h following hormonal stimulation and increased in embryos at the morula stage 6 h after hormonal stimulation (Figure 4B). In addition, Sanger sequencing revealed that long poly(A) tails of exogenous RNA injected into oocytes were not only removed but also uridylated after hormonal stimulation (Figure 4C, +1-MA 1 h). Thus, 1-MA stimulation shortened both endogenous and exogenous poly(A) tails of *Rps29* mRNA, followed by uridylation. Moreover, exogenous RNA carried U-rich non-canonical poly(A) tails in embryos at the morula stage (Figure 4C, +1-MA 6 h), as was observed for endogenous mRNA (Figure 2C, +1-MA 12 h). These results support the hypothesis that deadenylated and uridylated maternal RNAs following 1-MA stimulation were re-adenylated to generate U-rich non-canonical poly(A) tails in embryos at the morula stage. This re-adenylation signal is likely to be included in the 3′ UTR of *Rps29* mRNA as the exogenous mRNA carried only a 3′ UTR sequence and a unique sequence tag.

The targeted TAIL-Seq showed that the frequency of uridine additions at 3′ ends of injected *Rps29* mRNA increased following hormonal stimulation (Figure 4D, 2–11 h (+) 1-MA), whereas no increase was observed in unstimulated oocytes (Figure 4D, −1-MA 2 h, −1-MA 11 h). The frequency of non-A residues in the 5′ region was approximately 40% in exogenous *Rps29* mRNA tails from embryos at the morula and blastula stages (Figure 4E, left panel: +1-MA 5 h, +1-MA 11 h), which was comparable to that of endogenous *Rps29* mRNA tails (Figure 3C, left panel, blastula). However, the 3′ region contained more non-A than endogenous A residues (Figure 4E, right panel: +1-MA 5 h, +1-MA 11 h; Figure 3C, blastula), likely due to shorter tails of the injected RNAs (Figure 4B, right panel, +1-MA, 6 h) compared to those of endogenous mRNAs, which may contain U-rich sequences without poly(A) tails. The frequency of uridine was higher than that of guanine or cytosine in exogenous non-canonical poly(A) tails (Figure 4F).

### 3.5. AAUAAA Cleavage Recognition Site and Polyadenylation Specificity Factor (CPSF) Are Not Required for Re-Polyadenylation

Sanger sequencing revealed that some exogenous *Rps29* mRNAs in embryos at the morula stage or endogenous *Rpl27a* mRNAs in embryos at the blastula stage did not contain the polyadenylation signal (PAS) ‘AAUAA’ (Figure 4C, morula, lane 7; Appendix A blastula, lanes 4 and 9), suggesting that they were polyadenylated even after trimming of the 3′ UTR, including PAS. To confirm that PAS is not required for re-polyadenylation at morula or blastula stages, ΔPAS RNA was synthesized in which the U of the AAUAAA was mutated to G (Figure 5A). When this construct was injected into oocytes, followed by hormonal stimulation and fertilization, ΔPAS RNA behaved similarly during development to wild-type RNA carrying PAS (Figure 5B,C). The targeted TAIL-Seq of injected ΔPAS mRNA showed that the modification of the 3′ terminal region occurred similarly to that in the mRNA carrying PAS (Appendix A).

### 3.6. Non-Canonical Poly(A) Tailed mRNA Is Translationally Active

To investigate whether a non-canonical(A) tail enhances the translational activity of mRNAs, we applied a protein expression system for immature starfish oocytes injected with starfish *SGK* mRNA [28]. As the anti-sfSGK antibody was more sensitive than the anti-FLAG tag antibody, sfSGK was used as the protein tag. Four types of *sfSGK* mRNAs were synthesized: those carrying a long poly(A) tail (>100 nt), those carrying a short poly(A) tail (4 nt), and those carrying each of the two types of non-canonical poly(A) tails from blastula *Rps29* cDNA clones encoding nc1 and nc2 (Figure 6A). Western blot analysis of the oocytes injected with each mRNA to evaluate translational activity revealed that mRNAs with long poly(A) tails and nc1 tails supported substantial expression of exogenous SGK protein (Figure 6B). These results lead us to suggest that the A-rich poly(A) region (31 nt) in non-canonical poly(A) tails of nc1 mRNA exhibited translational activity comparable to those of long poly(A) tail (>100 nt). Furthermore, when reporter mRNA inserted between the 5′ and 3′ UTR of the 40S *Rps29* with a canonical poly(A) tail was synthesized and injected into stimulated oocytes, followed by fertilization (Figure 6C), embryos showed an increase in translational activities at the blastula stage (Figure 6D). As changes in the exogenous mRNA poly(A) tail length in injected embryos mimicked those in endogenous *Rps29* mRNA Figure 6E and Figure 2B), these results support the hypothesis that deadenylated and uridylated *Rps29* mRNAs are recycled, re-polyadenylated, and translated to produce ribosomal proteins at the blastula stage.

## 4. Discussion

This study used invertebrate starfish to investigate 3′-terminal modification in maternal mRNAs of ribosomal proteins and *cyclin B* during oocyte maturation and embryonic development. Starfish *cyclin B* mRNA carrying a long canonical poly(A) tail was deadenylated and uridylated at the blastula stage, followed by decay, indicating that uridylation of *cyclin B* mRNA promotes mRNA degradation in starfish embryos, as was previously demonstrated in vertebrate embryos [4,9] (Figure 7A, upper panel, #5). Conversely, *Rps29* and *Rpl27a* mRNAs producing starfish ribosomal proteins in oocytes did not decay even when they were deadenylated and uridylated following hormonal stimulation or fertilization (Figure 7A, upper panel, #1). In addition, these deadenylated and uridylated mRNAs were re-polyadenylated, forming translationally active long poly(A) tails at the blastula stage (Figure 7A, upper panel, #2). Thus, uridylated short poly(A) tails in starfish can have two fates: degradation (Figure 7A, #5) or poly(A) elongation (Figure 7A, #2 and #3), indicating that uridylation in a stably inactive “poised” state functions as a bidirectional signpost towards formation of unstable mRNAs for degradation or stably active mRNAs for translation (Figure 7A, lower panel; and Figure 7B). To the best of our knowledge, this is the first report demonstrating the re-adenylation of uridylated mRNAs to achieve secondary translational activity for mRNA recycling.

Mechanisms underlying the decision-making process for degradation or re-polyadenylation of uridylated mRNAs in starfish embryos still need to be fully elucidated. However, our results indicate that 3′ UTR is likely to be involved in the choice of fate and uridylation, as 3′ UTRs of the ribosomal protein mRNAs and *cyclin B* mRNA reproduced pattern of degradation or re-polyadenylation of endogenous mRNAs at morula, blastula, and gastrula stages (Figure 1 and Figure 4). Upon resumption of meiosis in *Xenopus*, mouse, and starfish oocytes, cytoplasmic polyadenylation of many maternal mRNAs, including *cyclin B*, is regulated by 3′ UTRs containing the PAS AAUAAA and CPE, which are bound by CPSF and CPEB, respectively [22,42,43,44]. Subsequently, Gld2, which interacts with CPSF and CPEB, elongates poly(A) tails [24,45,46]. Alternatively, mRNAs that do not contain CPE or PAS are deadenylated after hormonal stimulation in both amphibian and mouse oocytes [21,47,48]. Similarly, maternal mRNAs of starfish ribosomal proteins, *Rps29* and *Rpl27a*, which do not contain CPE, were deadenylated after hormonal stimulation (Figure 2 and Appendix A). Moreover, re-polyadenylation of the *Rps29* mRNA occurred even in endogenous mRNA trimmed from 3′ ends to the site of PAS, and in the exogenous 3′ UTR carrying a mutation in PAS (Appendix A blastula, lanes 4 and 9; Figure 4C, morula, lane 7), suggesting that Gld2 may not be involved in non-canonical poly(A) elongation.

Deadenylated starfish *cyclin B* mRNA was uridylated at the blastula stage, followed by decay, suggesting that terminal uridylyltransferases, such as mammalian TUT4/7 [4,12], may be involved (Figure 7A, #4). In *Xenopus* embryos, degradation of *cyclin B2* mRNA depends on 3′ UTR [38], which contains microRNA-427 (miR-427) target sequence, with the zygotic expression of miR-427 inducing *cyclin B2* mRNA deadenylation [49]. Similarly, decay of starfish *cyclin B* mRNA may also be induced by microRNA, as duplex formation of morpholino oligonucleotides mimicking interaction of microRNA with mRNA 3′ UTR terminus degrades the mRNA in starfish [32].

In *Xenopus*, many maternal mRNAs are not decayed immediately following deadenylation but are stable until the blastula stage, several hours after fertilization [14,15,16,17,50]. While we still need to determine how *Xenopus* maternal mRNAs with short poly(A) tails are stabilized after deadenylation, there is a possibility that uridylation of *Xenopus* maternal mRNAs does not necessarily cause decay, similar to results observed in starfish. Consistent with this, when we analyzed TAIL-Seq data of *Xenopus* embryos kindly provided by Hyeshik Chang and Narry Kim [4], we found that approximately 20% of the *Xenopus* 40S *rps29* mRNAs were modified to be uridylated at stage 5 (Appendix A). Thus, uridylated *Xenopus* mRNAs, as demonstrated in starfish, may exhibit stability. If this is the case, it would support the existence of a stable inactive state of uridylated mRNAs shared by both vertebrates and invertebrates.

Non-A residues are observed more frequently near the 5′ end of poly(A) tails in mouse GV oocytes, as shown by the poly(A) inclusive RNA isoform sequencing (PAIso−Seq) [51]. Similarly, approximately 30% of non-A residues were predominantly distributed in the 5′ region of non-canonical poly(A) tails of mRNA of the ribosomal protein in starfish embryos (Figure 2C and Figure 3C), suggesting that both animals may utilize similar modification systems. Although enzymes involved in the 5′ end modifications of mouse oocytes still need to be determined, mammalian TENT4A/4B [52] can mediate mixed tailing of adenylation and guanylation to stabilize mRNAs. Thus, the starfish homolog of TENT4 may generate 5′ modifications in non-canonical poly(A) tails. In addition, TENT5 families mediate cytoplasmic polyadenylation of collagen mRNAs required for bone mineralisation [53] and immunoglobulin mRNA stabilization to enhance expression [54]. These families may also participate in non-canonical polyadenylation in starfish.

In this study, we propose that uridylated mRNAs have a stable inactive state in starfish. Although it remains to be confirmed whether stable mRNAs with uridylated tails exist in other organisms, suggestive results from *Xenopus* (Appendix A) and the presence of uridylated mRNAs in various cells [10,20] indicate that additional waiting phases are required to recruit molecules for degradation or possibly re-elongation of mRNA tails. Further research is required to improve our understanding of recycling [55] and re-elongation of uridylated mRNAs in other developmental stages of starfish embryos, as well as across a broader range of animals, including marine organisms.

## 5. Conclusions

In starfish oocytes, uridylated maternal cyclin B mRNAs remain stable and are polyadenylated to enable translation following hormonal stimulation. These mRNAs are deadenylated and re-uridylated, ultimately undergoing decay after the blastula stage. In contrast, maternal ribosomal protein mRNAs, such as *Rps29* and *Rpl27a*, are polyadenylated and translationally active prior to hormonal stimulation. Following stimulation, these mRNAs become deadenylated, uridylated, and translationally inactive. However, at the morula stage, the uridylated maternal ribosomal protein mRNAs are re-polyadenylated, restoring their translational activity. These findings lead us to suggest that uridylated mRNAs in starfish adopt a poised state, allowing them to be either recycled for translation or targeted for decay.

## Figures and Tables

**Figure 1 biomolecules-14-01610-f001:**
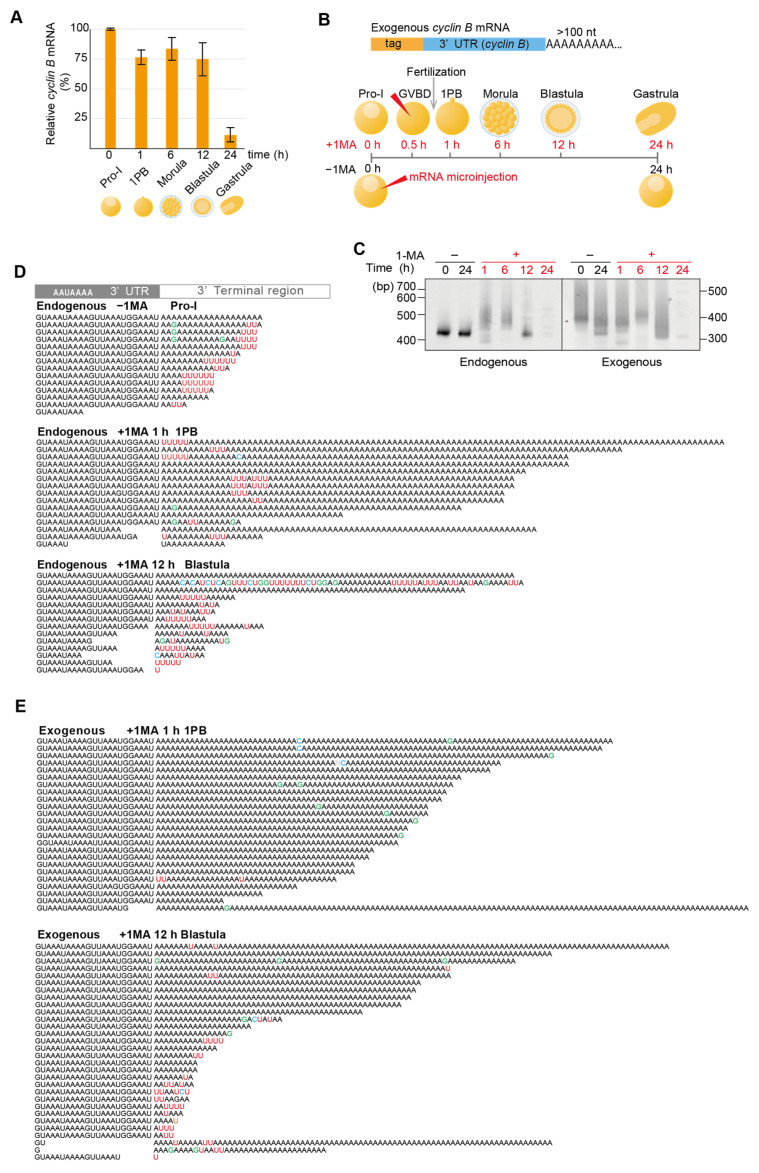
*Cyclin B* mRNA uridylation and degradation. (**A**) Relative mRNA expression levels for *cyclin B* in starfish oocytes and embryos. Total RNAs were purified at the indicated time and relative quantification of *cyclin B* mRNA expression was conducted using RT-qPCR (median ± SEM) (*n* = 3). (**B**) Experimental scheme for the microinjection of artificially tagged *cyclin B* mRNA. Tag-labeled *cyclin B* mRNA was synthesized and injected (red arrowheads) into oocytes at 0.5 h after 1-MA treatment (+1-MA) or into oocytes without the 1-MA treatment (−1-MA). Oocytes (+1-MA) were inseminated before the first polar body formation, and the total RNAs were purified at the indicated time. (**C**) Tail length measurements for *cyclin B* mRNA. At the indicated time, the total RNAs were purified, and a TGIRT template-switching reaction was performed (Appendix A). RT-PCR was conducted using the 3′ adaptor reverse primer and *cyclin B*-specific forward primer. The PCR products were then subjected to polyacrylamide gel electrophoresis and visualized using SYBR-Green I staining. The left and right panels show changes in the tail lengths of endogenous *cyclin B* mRNA and exogenously microinjected *cyclin B* mRNAs, respectively. (**D**) Sanger sequencing results of the 3′ terminal region of cDNA from endogenous maternal *cyclin B* mRNA of oocytes at Pro-I without 1-MA treatment [Endogenous −1-MA], from stimulated oocytes following first polar body formation (1 h after 1-MA treatment) [Endogenous +1-MA 1 h], and from embryos at the blastula stage (12 h after 1-MA treatment) [Endogenous +1-MA 12 h]. The mean number ± SE of uridine residues in the five nucleotides from the 3′ end was 0.0 ± 0.0 for [Endogenous −1-MA], 2.6 ± 0.5 for [Endogenous +1-MA 1 h], and 1.4 ± 0.3 for [Endogenous +1-MA 12 h]. A Tukey HSD test revealed that the mean number in the [Endogenous −1-MA] group was significantly lower than those in the other two groups (*p* < 0.05). (**E**) Sequencing results of the 3′ terminal region of exogenous *cyclin B* mRNA from stimulated oocytes following first polar body formation (1 h after 1-MA treatment) [Exogenous +1-MA] and from embryos at the blastula stage (12 h after 1-MA treatment) [Exogenous −1-MA].

**Figure 2 biomolecules-14-01610-f002:**
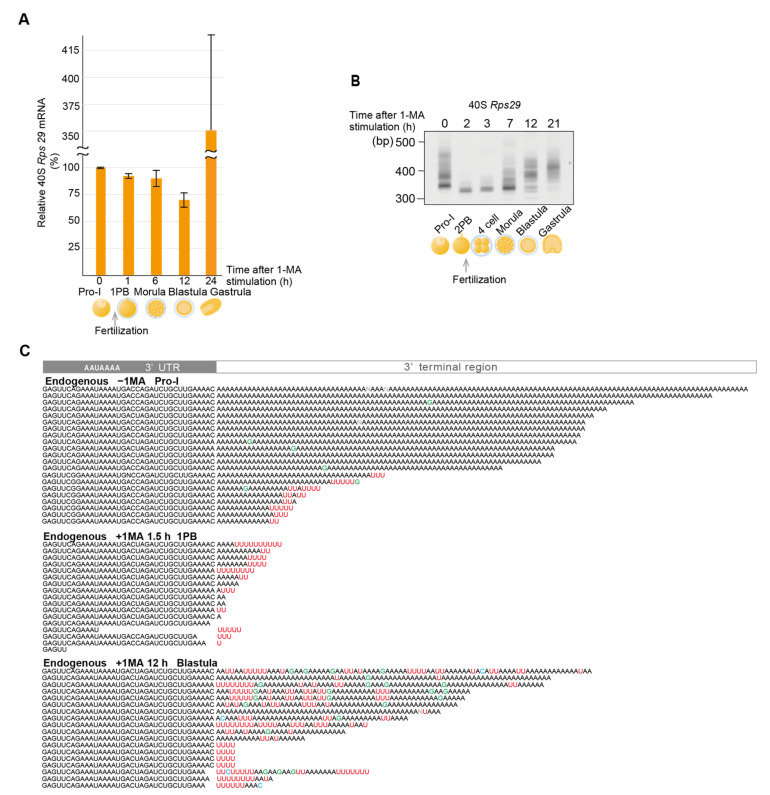
Deadenylation, uridylation, and non-canonical poly(A) elongation of the mRNA for ribosomal protein *Rps29*. (**A**) Relative levels of mRNA expression for the ribosomal protein in starfish oocytes and embryos. Total RNAs were purified at the indicated time and relative quantification of mRNA expression for the ribosomal protein *Rps29* was performed using RT-qPCR (median ± SEM) (*n* = 3). (**B**) Measurement of the *Rps29* mRNA tail lengths. Oocytes stimulated with 1-MA were inseminated following second polar body formation. At the indicated time before or after 1-MA stimulation, total RNAs were purified, and adaptor ligation was performed (Appendix A). RT-PCR was conducted using the 3′ adaptor reverse primer and *Rps29*-specific forward primer. The PCR products were then subjected to polyacrylamide gel electrophoresis and visualized using SYBR-Green I staining. Similar results were obtained for three animals. (**C**) Sequencing results of the 3′ terminal region of cDNA of endogenous maternal *Rps29* mRNA from oocytes at Pro-I without 1-MA treatment [Endogenous −1-MA], stimulated oocytes after the first polar body formation (1.5 h after 1-MA treatment) [Endogenous +1-MA 1.5 h], and embryos at the blastula stage (12 h after 1-MA treatment) [Endogenous +1-MA 12 h]. The mean number ± SE of uridine residues in the five nucleotides from the 3′ end was 0.0 ± 0.0 for [Endogenous −1-MA] (poly(A) length > 40 residues), 1.9 ± 0.5 for [Endogenous +1-MA 1.5 h] and 0.4 ± 0.2 for [Endogenous +1-MA 12 h] (poly(A) length > 40 residues). A Tukey HSD test revealed that the mean number in the [Endogenous −1-MA] group was significantly lower than the [Endogenous +1-MA 1.5 h] group (*p* < 0.05).

**Figure 3 biomolecules-14-01610-f003:**
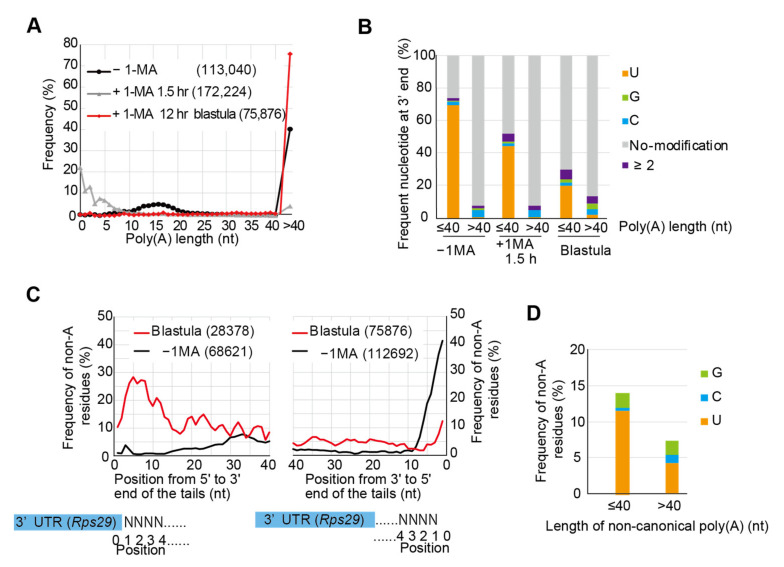
Targeted TAIL-Seq of *Rps29* mRNA. (**A**) Distribution of poly(A) tail lengths of *Rps29* mRNA from oocytes without 1-MA treatment (−1-MA), with 1-MA treatment for 1.5 h (+1-MA 1.5 h), and with 1-MA treatment followed by insemination to obtain blastulae at 12 h. Relative frequencies (Y-axis, %) were calculated by dividing the number of detected reads that have indicated poly(A) lengths by the total number of reads that have poly(A) tails. Frequencies of poly(A) tail length > 40 nucleotides are plotted on the right side. The number of reads is shown in parentheses. (**B**) Relative frequencies of the most frequent nucleotide with additional modifications at the 3′ end of the *Rps29* mRNA. Using each *Rps29* mRNA read, the most frequent nucleotides, such as U, G, and C, were determined. Relative frequencies (Y-axis, %) were calculated by dividing the number of reads with the most frequent nucleotides by the total number of reads with the indicated lengths of the poly(A) tails. mRNAs with tail lengths of ≤40 nucleotides and >40 nucleotides were compared in the oocytes with or without 1-MA and at the embryonic stages of development. No modification, neither poly(A) tail nor additional modifications, were present at the end of the poly(A) tails. ≥2; two or more nucleotides comprised the most frequent nucleotides in the mRNA. (**C**) Distribution of the relative frequencies of non-A residues in the *Rps29* mRNA tails from blastulae and unstimulated oocytes. At the indicated position of the tails, the relative frequencies of the non-A residues (Y-axis, %) were calculated by dividing the number of reads carrying non-A residues by the total number of reads. The distribution of frequencies for the non-A residues is shown at the indicated position in the tails from 5′ to 3′ (left panel) and from 3′ to 5′ (right panel). The numbers of reads are shown in parentheses. (**D**) Relative frequencies of non-A residues in the non-canonical poly(A) tails of *Rps29* mRNA. The relative frequencies of the non-A residues (Y-axis, %) were calculated by dividing the number of each non-A residue (G, C, and U) in the tails of all reads by the number of tail lengths for all reads that have indicated lengths of poly(A) tails. mRNAs with tail lengths of ≤40 nucleotides and >40 nucleotides are compared at the blastula stage.

**Figure 4 biomolecules-14-01610-f004:**
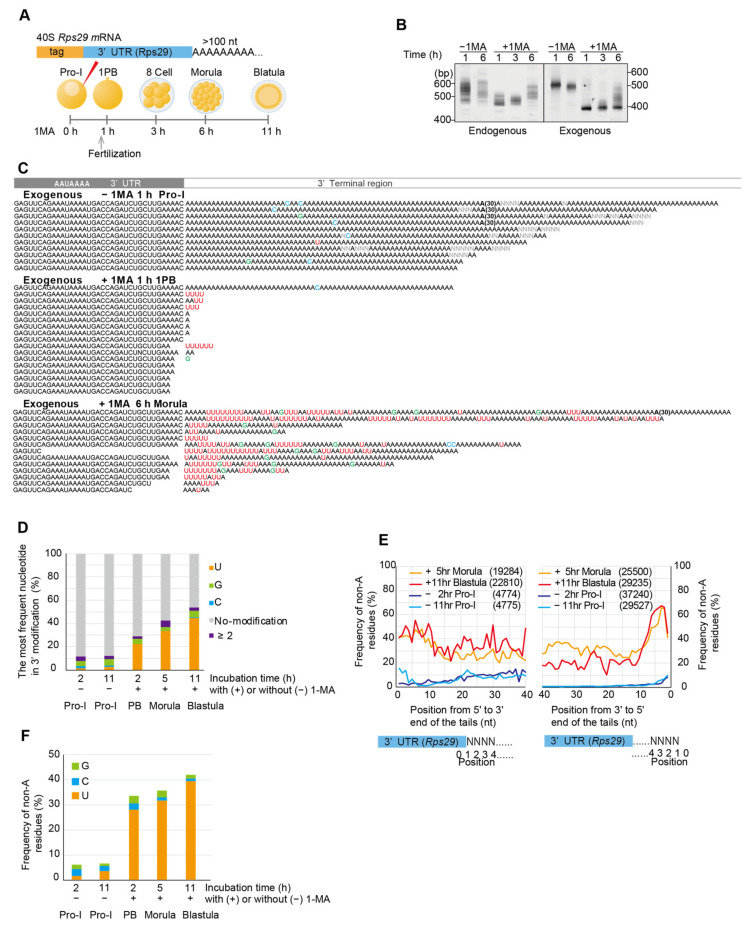
Deadenylation, uridylation, and non-canonical poly(A) elongation of the mRNA for exogenous ribosomal protein *Rps29*. (**A**) Experimental scheme. Tag-labeled *Rps29* mRNAs carrying long poly(A) tails were injected into oocytes with or without 1-MA treatment. Total RNAs were purified at the indicated times, whereas stimulated oocytes at 1 h and unstimulated oocytes were not inseminated. (**B**) Measurement of the tail length of endogenous and exogenous *Rps29* mRNA. RT-PCR was conducted using the 3′ adaptor reverse primer and *Rps29* mRNA-specific forward primer. The PCR products were subjected to polyacrylamide gel electrophoresis and SYBR-Green I staining. Similar results were obtained for two animals. (**C**) Sequencing results of exogenous *Rps29* mRNA from oocytes without 1-MA treatment [−1-MA], oocytes at 1 h after 1-MA treatment [+1-MA 1 h], and embryos at the morula stage (6 h after 1-MA treatment) [+1-MA 6 h]. ‘(30)’ in the sequences indicates ‘AA…AA’ containing 30 nucleotides. (**D**) Relative frequencies of the most frequent nucleotides in the additional modifications at the 3′ end of exogenous *Rps29* mRNA. Relative frequencies (Y-axis, %) were calculated by dividing the number of reads carrying the most frequent nucleotides by the total number of reads from oocytes and embryos at the indicated time. No modification, neither poly(A) tail nor additional modifications were present. “≥2”; two or more nucleotides comprised the most frequent nucleotides. (**E**) Distribution of the relative frequencies of non-A residues in tails of exogenous *Rps29* mRNA from oocytes and embryos. At the indicated position of the tails, the relative frequencies of the non-A residues (Y-axis, %) were calculated by dividing the number of reads carrying the non-A residues by the total number of reads. The distribution of frequencies of non-A residues is shown at the indicated position in tails from 5′ to 3′ (left panel) and from 3′ to 5′ (right panel). The numbers of reads are shown in parentheses. Yellow, morulae (5 h after 1-MA treatment). Red, blastulae (11 h after 1-MA treatment). Purple, Pro-I oocytes without 1-MA stimulation (2 h after injection). Blue, Pro-I oocytes without 1-MA stimulation (11 h after injection). (**F**) Relative frequencies of non-A residues in poly(A) tails of exogenous *Rps29* mRNA. The relative frequencies of the non-A residues (Y-axis, %) were calculated by dividing the number of each non-A residue (G, C, and U) in the tails of all reads by the number of tail lengths of all reads.

**Figure 5 biomolecules-14-01610-f005:**
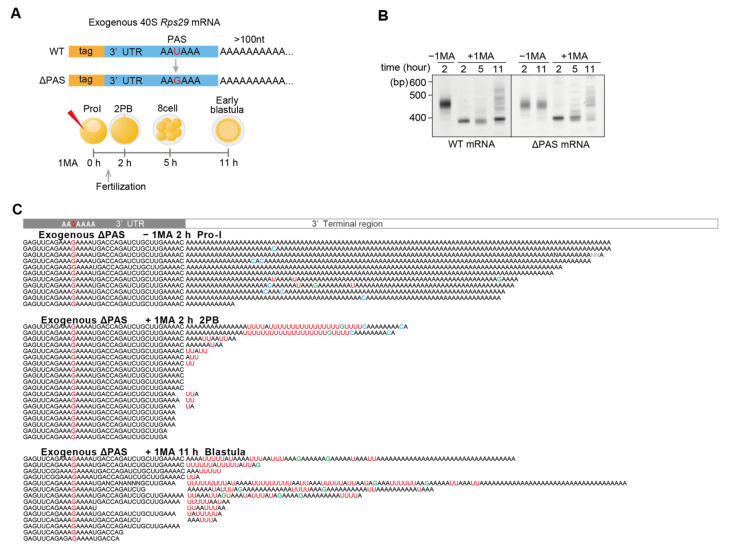
Deadenylation, uridylation, and non-canonical poly(A) elongation of the mRNA for exogenous ribosomal protein ∆ PAS *Rps29*. (**A**) Experimental scheme. Tag-labeled wild-type and ∆ PAS *Rps29* mRNAs were injected into oocytes. At the indicated time, the total RNA was purified. (**B**) Measurement of the *Rps29* mRNA tail length. Total RNA was subjected to a TGIRT template-switching reaction. RT-PCR was conducted using the 3′ adaptor reverse primer and Tag-specific forward primer. The PCR products were then subjected to polyacrylamide electrophoresis and visualized using SYBR-Green I staining. The left and right panels show the changes in tail lengths for the exogenous wild-type and ∆ PAS *Rps29* mRNAs, respectively. Similar results were obtained for two animals. (**C**) Sequencing results of the 3′ terminal region of exogenous ∆ PAS *Rps29* mRNA purified from oocytes at Pro-I without 1-MA treatment at 2 h following injection of the mRNA [−1-MA 2 h], stimulated oocytes at 2 h following 1-MA treatment [+1-MA 2 h], and blastulae (11 h after 1-MA treatment) [+1-MA 11 h].

**Figure 6 biomolecules-14-01610-f006:**
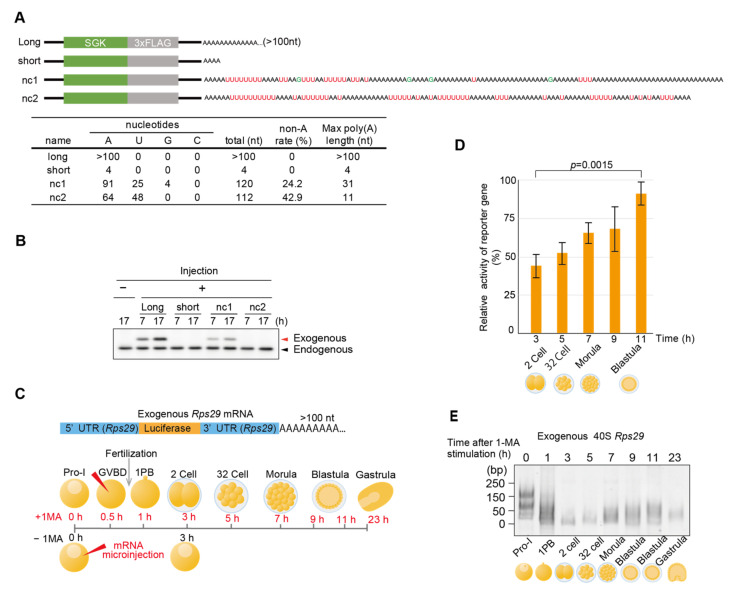
Translational activities of the non-canonical poly(A) tails. (**A**) Experimental scheme. Tag(SGK)-labeled 40S *Rps29* mRNAs with canonical long poly(A) tails, short poly(A) tails, and non-canonical long and short poly(A) tails were synthesized and injected into oocytes without the 1-MA treatment. The table shows the number of A, T, G, and C nucleotides in the tails. (**B**) Western blotting of oocytes injected with mRNA carrying an *SGK* and non-canonical poly(A) tails of the 40S *Rps29* mRNA. At the indicated time after the injection of exogenous mRNA, oocytes were treated with a sample buffer, followed by polyacrylamide gel electrophoresis and Western blotting using an anti-SGK antibody. Arrowheads (endogenous) indicate endogenous SGK in oocytes. Similar results were obtained for 3 animals. (**C**) Experimental scheme. Reporter luciferase mRNA between the 5′ and 3′ UTR of 40S *Rps29* with a canonical long poly(A) tail was injected into oocytes with or without the 1-MA treatment. D, E. 1-MA-stimulated oocytes were inseminated after GVBD to start embryonic development and used to determine the reporter activities (**D**) (mean ± SE) (*n* = 3) and the length of poly(A) tails (**E**). To calculate the relative activity of translation, the luciferase activity of the unstimulated oocytes at 3 h after injection of the reporter mRNA was considered to be 100%, and Student’s t-test was used to determine the significance between the results observed after 3 and 11 h (**D**). Similar results were obtained for two animal models (**D**,**E**).

**Figure 7 biomolecules-14-01610-f007:**
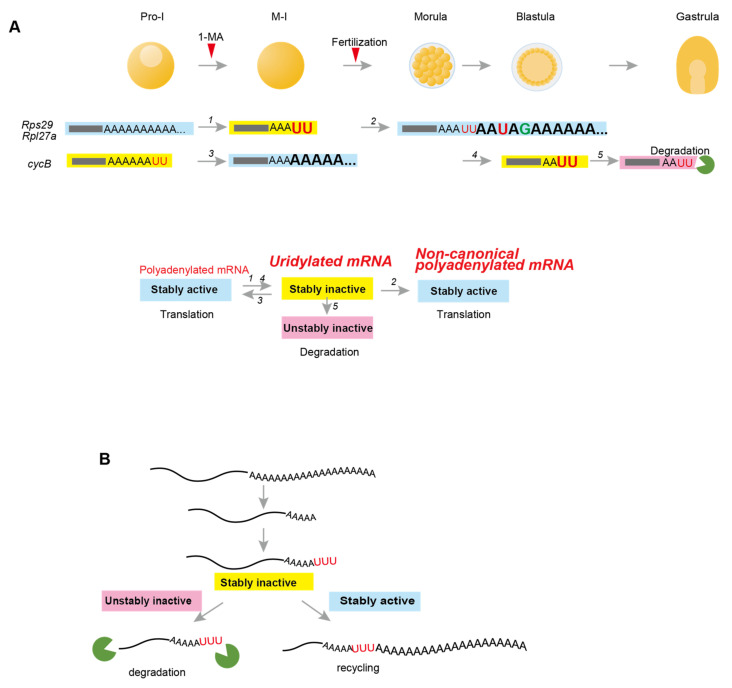
Summary of the findings and proposed model. (**A**) Maternal mRNA deadenylation, uridylation, and non-canonical poly(A) elongation in starfish oocytes and embryos. Upper panel: Before hormonal stimulation of 1-MA, oocytes at Pro-I contain maternal ribosomal protein mRNAs (*Rps29* and *Rpl27a*) carrying long poly(A) tails and *cyclin B* mRNAs with uridylated short poly(A) tails. Following the resumption of meiosis of oocytes undergoing nuclear division of germinal vesicle breakdown, long poly(A) tails of ribosomal protein mRNAs are deadenylated and uridylated (newly added nucleotides are shown in larger font size). Some uridine residues of *cyclin B* mRNA are trimmed [26], followed by poly(A) elongation. After fertilization, uridylated short poly(A) tails of ribosomal proteins in morulae are re-elongated, forming non-canonical poly(A) tails. At the blastula stage, *cyclin B* mRNAs are deadenylated and uridylated. They are then degraded before gastrulation. Lower panel: Upper panel summarization. The arrow numbers correspond to those in the upper panel. Uridylated mRNAs are stable but inactive for translation. Canonical or non-canonical polyadenylation renders inactive mRNAs stably active for translation. Alternatively, uridylated mRNAs become unstable and inactive, followed by decay, as observed in other animals. (**B**) Proposed model for uridylation in starfish. Uridylated mRNAs are required to determine mRNA fate: destruction or recycling.

## Data Availability

The MiSeq sequencing data generated in this study were deposited under BioProject PRJDB9545 (https://www.ncbi.nlm.nih.gov/bioproject/?term=PRJDB9545, accessed on 13 September 2019). The accession numbers of the mRNAs of Rps29 (40S ribosomal protein) and Rpl27a (60S ribosomal protein) are LC535328 and LC535329, respectively.

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
