# Peer review of "Recycling of Uridylated mRNAs in Starfish Embryos"

_biomolecules, 2024, doi:10.3390/biom14121610_

Round 1

Reviewer 1 Report

Comments and Suggestions for Authors

This study by Yamazaki et al investigates the fate of uridylated mRNAs in starfish, revealing that some uridylated maternal mRNAs, such as cyclin B and ribosomal protein mRNAs, can be re-polyadenylated and translationally reactivated. Cyclin B mRNAs remain stable and are re-polyadenylated after hormonal stimulation to resume meiosis, but they decay after the maternal-to-zygotic transition (MZT). In contrast, ribosomal protein mRNAs undergo re-polyadenylation at the morula–blastula stage, becoming translationally active. These findings suggest that uridylated mRNAs can exist in a poised state, enabling selective recycling or decay, and provide insights into the regulatory mechanisms of mRNA modifications during early starfish development.

The microinjection experiments are interesting, but how physiological relevant are the in vitro synthesis RNA concentration? The concentration of infected RNA varied (10-fold 10-100 ng/ul) and the injection volume varies two-fold, so how is this controlled for in the experiments? This is especially important for the analysis of translational efficacy (Figure 6). Also does the injected RNA accurately endogenous mRNA levels, and is there an issue in diluting regulator factors if too much exogenous RNA is added? 

A limitation of the study is the restricted number of RNAs examined, so it makes is harder to extrapolate the finding to other RNAs. Scaling up the scale TAIL-Seq to look at additional target mRNAs would improve the significance of the results.

The TAIL-Seq data of analysis of Xenopus maternal mRNA data finds only 20% of 40S rps29 is uridylated. It is not clear how does this compare with the starfish data.

Minor comments:

The figures are well constructed but the much of the key detail is hard to look at due to the small size of fonts/images. It would be good to increase the readability of the figures.

Figure 4.  Outline the PAS signal sequence in the alignment.

The paragraph in the discussion about suggesting a possible similarity between mammalian and starfish LARP1 is very speculative. What is the level of homology between LARP1 in the two species and what is known about the expression of starfish LARP1? This needs to be fleshed out further if this line of argument is to be included.

Author Response

We would like to thank the reviewer for their constructive comments and suggestions. Please find the detailed responses below and the corresponding revisions/corrections highlighted in the re-submitted files.

Comments 1: 

This study by Yamazaki et al investigates the fate of uridylated mRNAs in starfish, revealing that some uridylated maternal mRNAs, such as cyclin B and ribosomal protein mRNAs, can be re-polyadenylated and translationally reactivated. Cyclin B mRNAs remain stable and are re-polyadenylated after hormonal stimulation to resume meiosis, but they decay after the maternal-to-zygotic transition (MZT). In contrast, ribosomal protein mRNAs undergo re-polyadenylation at the morula–blastula stage, becoming translationally active. These findings suggest that uridylated mRNAs can exist in a poised state, enabling selective recycling or decay, and provide insights into the regulatory mechanisms of mRNA modifications during early starfish development.

The microinjection experiments are interesting, but how physiological relevant are the in vitro synthesis RNA concentration? The concentration of infected RNA varied (10-fold 10-100 ng/ul) and the injection volume varies two-fold, so how is this controlled for in the experiments? This is especially important for the analysis of translational efficacy (Figure 6). Also does the injected RNA accurately endogenous mRNA levels, and is there an issue in diluting regulator factors if too much exogenous RNA is added?

Responses 1:

As you suggested, exogenous mRNA injection might potentially interfere with the regulation of endogenous mRNA modifications. However, we consider this possibility negligible based on the following observations:

  1. Deadenylation and degradation of cyclin B endogenous mRNA following hormone stimulation (Fig. 1) were fully recapitulated by the injection of synthetic cyclin B mRNA, demonstrating that exogenous mRNA does not disrupt the regulatory mechanisms of endogenous mRNA.
  2. Similarly, the deadenylation and re-adenylation of ribosomal protein endogenous mRNA (Fig. 2) were reproduced by the injection of synthetic ribosomal protein mRNA (Fig. 4), further confirming that the injection of exogenous mRNA does not interfere with the regulation of endogenous mRNA.
  1. The polyadenylation pattern of endogenous mRNA (Fig. 4B) closely resembles that of exogenous mRNA (Fig. 6E), which correlates with increased translation. These findings support the hypothesis that polyadenylation of endogenous mRNA similarly enhances its translation.

Comments 2:

A limitation of the study is the restricted number of RNAs examined, so it makes is harder to extrapolate the finding to other RNAs. Scaling up the scale TAIL-Seq to look at additional target mRNAs would improve the significance of the results.

Responses 2:

While our preliminary experiments indicate that other mRNAs also undergo uridylation, scaling up Tail-Seq analysis requires significant time and manpower. We greatly appreciate your suggestion and plan to address these experiments in detail in future studies, with the findings to be reported in subsequent manuscripts.

Comments 3:

The TAIL-Seq data of analysis of Xenopus maternal mRNA data finds only 20% of 40S rps29 is uridylated. It is not clear how does this compare with the starfish data.

Responses 3:

The analysis of Xenopus data was incorporated to indicate that mechanisms of uridylation-dependent mRNA regulation, similar to those in starfish, may also be partially active (approximately 20%) in Xenopus. To make this relationship clearer, we revised the text as follows:

Before:

Thus, some Xenopus mRNAs may be stable even when uridylated, suggesting the presence of a stable inactive state of uridylated mRNAs in both vertebrates and invertebrates.

After (Line 811):

Thus, uridylated Xenopus mRNAs, as demonstrated in starfish, may exhibit stability. If this is the case, it would support the existence of a stable inactive state of uridylated mRNAs shared by both vertebrates and invertebrates.

Comments 4:

Minor comments:

The figures are well constructed but the much of the key detail is hard to look at due to the small size of fonts/images. It would be good to increase the readability of the figures.

Figure 4.  Outline the PAS signal sequence in the alignment.

Responses 4:

As suggested, we have adjusted the font size of the outlined PAS signal in Figure 4, as well as in Figures 1, 2, and 5.

Comments 5:

The paragraph in the discussion about suggesting a possible similarity between mammalian and starfish LARP1 is very speculative. What is the level of homology between LARP1 in the two species and what is known about the expression of starfish LARP1? This needs to be fleshed out further if this line of argument is to be included.

Responses 5:

 As indicated, we removed the following sentences: “The La-related protein 1 (LARP1), which recognises the UUU-3’-OH of the terminal motif of transcripts synthesised by RNA polymerase III in the nuclei [53], can control the translation and stability of mammalian ribosomal protein mRNAs in the cytoplasm [54,55]. There is therefore also a possibility that the stability of uridylated ribosomal protein mRNAs in the cytoplasm of early embryos in starfish may also be regulated by molecules such as LARP1.”

Reviewer 2 Report

Comments and Suggestions for Authors

The present study investigated the dynamics of uridylated mRNAs in starfish embryos with respect to their stability, re-polyadenylation, and translational activation during their development. This study is an in-depth investigation into mRNA tail modifications and therefore constitutes a critical contribution to our knowledge of post-transcriptional regulation during early embryogenesis. These authors now provide a striking set of findings to support the novel observation that uridylated mRNAs can be recycled and reactivated, in contrast to the dogma of predicting uridylation as a signal for degradation.

This paper is well organized, comprehensive, and replete with robust methodologies. Its strengths include novelty in the use of tail-seq and microinjection techniques, and the clarity of data presentation. On the contrary, weaknesses identified in the manuscript are discussions, especially where integration of existing literature could be more effective or limitations of potential problems being discussed more clearly.

Some comments that the authors should consider are enumerated below:

1.     Introduction: It provides a good background to a large extent about the importance of mRNA tail modifications in cells. The research question is well-interpreted; however, for contextualization–how starfish systems relate to wider biological processes–an explanation would be helpful. Relevant references have been cited; however, completeness could be achieved by citing a recent review about the mechanisms that regulate the mRNA tail in other non-vertebrate systems.

2.     Methods: The methodology section was extensive; therefore, it can be reproduced. Most techniques, such as microinjection, tail-seq, and RT-PCR, are described without affecting text clarity. Ethical considerations were met, as the study was based on the ARRIVE guidelines. However, while the methods applied in data collection are sufficient, a short rationale for certain time points of experiments should be provided, for example, stimulation at 1 h. A flowchart summarizing the workflow is required for better clarity.

3.     Results: Well-presented results are presented with appropriate figures and tables. The presentation of data, especially the results obtained from electrophoresis and sequencing, is clear. The section.  exceptionally well at substantiating the main conclusions. However, some of the data, such as the variation in the uridylation patterns of cyclin B and ribosomal protein mRNAs, could be subjected to a different statistical approach to emphasize their significance.

4.     Discussion: This discussion places the findings well into the existing literature and points toward the role of uridylation other than degradation. This manuscript could benefit from further consideration of the possible molecular mechanisms that account for the differences in stability among uridylated mRNAs. In addition, a short discussion of how these findings might inform future research on mRNA stability in other marine organisms would increase the scope of this study. The limitations section is superficial, with important biases, especially in the selection of particular stages in development.

5.     Conclusion: The conclusions were drawn to match the data presented. The authors amply express in the text that, in starfish, the uridylated mRNAs can be reclaimed for translational activity with the addition of a poly tail, thus providing an important insight into the topic at hand. This section can be further upgraded by suggesting studies that would confirm whether such post-transcriptional processes occur in other organisms.

6.     Major Flaws:

• The study did not consider the possible role of uridylation in non-maternal mRNAs, which would extend the generality of the findings.

• It should be better hypothesized how the mechanisms underpin why some uridylated mRNAs are re-polyadenylated.

7.     Minor Issues:

• Explain the rationale behind the choice of time points for hormonal stimulation, such as 1 and 6 h.

           Some sentences in the discussion can be simplified in a more direct manner. For example, from lines 411 to 415:.

           Some typing errors in Table S1 need to be corrected.

8.        The manuscript provides several issues of concern regarding the similarity indices, an overall similarity score of 75% that is above acceptable thresholds, and a single source-matched problem of 38% . A manuscript should generally have a below-30% overall similarity, as stated in the MDPI's guidelines for publication ethics, although the similarity to any single source should not be more than 15%. These scores require revision, involving substantial improvements, before consideration is made to publish the manuscript.

Author Response

We would like to thank the reviewer for their constructive comments and suggestions. Please find the detailed responses below and the corresponding revisions/corrections highlighted in the re-submitted files.

Comments 1:

Comments and Suggestions for Authors

The present study investigated the dynamics of uridylated mRNAs in starfish embryos with respect to their stability, re-polyadenylation, and translational activation during their development. This study is an in-depth investigation into mRNA tail modifications and therefore constitutes a critical contribution to our knowledge of post-transcriptional regulation during early embryogenesis. These authors now provide a striking set of findings to support the novel observation that uridylated mRNAs can be recycled and reactivated, in contrast to the dogma of predicting uridylation as a signal for degradation.

This paper is well organized, comprehensive, and replete with robust methodologies. Its strengths include novelty in the use of tail-seq and microinjection techniques, and the clarity of data presentation. On the contrary, weaknesses identified in the manuscript are discussions, especially where integration of existing literature could be more effective or limitations of potential problems being discussed more clearly.

Some comments that the authors should consider are enumerated below:

  1. Introduction: It provides a good background to a large extent about the importance of mRNA tail modifications in cells. The research question is well-interpreted; however, for contextualization–how starfish systems relate to wider biological processes–an explanation would be helpful.

Responses 1:

We believe that the starfish system offers significant advantages for studying maternal mRNA modifications due to its unique developmental characteristics. In starfish, new mRNA synthesis is entirely absent until the blastula stage, during the maternal-to-zygotic transition (MZT). This allows us to exclusively monitor maternal mRNA modifications from the one-cell stage without the confounding influence of newly synthesized mRNA. In contrast, in vertebrates, MZT occurs as early as the two-cell stage, complicating the determination of whether 3′ mRNA tail structures arise from newly transcribed nucleotides or modifications of pre-existing transcripts during development. To emphasize this advantage, we revised the manuscript as follows:

Before:

In the present study, we investigated the fate of maternal cyclin B and ribosomal protein mRNAs during starfish development. As a result of maternal-to-zygotic transition (MZT) [28], new mRNA synthesis does not occur until the blastula stage in starfish. Owing to this physiological block of transcription, we could trace the modifications in sole maternal mRNA tails during development and reveal that uridylated ribosomal protein mRNAs are re-polyadenylated and translationally reactivated at the stage of morula–blastula embryos, while uridylated cyclin B mRNAs decay after MZT.

After (Line 77-):

In the present study, we investigated the fate of maternal cyclin B and ribosomal protein mRNAs during starfish development. As a result of the maternal-to-zygotic transition (MZT), new mRNA synthesis does not occur until the blastula stage in starfish[29,30]. This long physiological block of transcription uniquely enables the tracing of sole maternal mRNA modifications throughout development. In vertebrates, however, MZT occurs as early as the two-cell stage[31], complicating the distinction between modifications of maternal transcripts and newly synthesized zygotic mRNA. Using the starfish system, we revealed that uridylated ribosomal protein mRNAs are re-polyadenylated and translationally reactivated at the morula–blastula stage, while uridylated cyclin B mRNAs decay after MZT.

Comments 2:

Relevant references have been cited; however, completeness could be achieved by citing a recent review about the mechanisms that regulate the mRNA tail in other non-vertebrate systems.

Responses 2:

As suggested, we added a recent review “Structure and function of molecular machines involved in deadenylation-dependent 5′-3′ mRNA degradation” by Zhao (2023) in the sentence (Line 41): “indicating that the deadenylation and uridylation of mRNA tails constitutes a con-served system of RNA degradation in eukaryotes [11,12].”

Comments 3:

Methods: The methodology section was extensive; therefore, it can be reproduced. Most techniques, such as microinjection, tail-seq, and RT-PCR, are described without affecting text clarity. Ethical considerations were met, as the study was based on the ARRIVE guidelines. However, while the methods applied in data collection are sufficient, a short rationale for certain time points of experiments should be provided, for example, stimulation at 1 h. A flowchart summarizing the workflow is required for better clarity.

Responses 3:

As suggested, we added the flowchart summarizing the workflow in Fig. S1F, and added the following sentence in the section of Microinjection: (Line 176) Injection timing and workflow are shown (Fig. S1F).

Comments 4:

Results: Well-presented results are presented with appropriate figures and tables. The presentation of data, especially the results obtained from electrophoresis and sequencing, is clear. The section. exceptionally well at substantiating the main conclusions. However, some of the data, such as the variation in the uridylation patterns of cyclin B and ribosomal protein mRNAs, could be subjected to a different statistical approach to emphasize their significance.

Responses 4:

As suggested, we revised Figure 1 D legend as follows:

Before:

Sanger sequencing results of the 3’ terminal region of cDNA from endogenous maternal cyclin B mRNA of oocytes at Pro-I without 1-MA treatment [Endogenous −1-MA], from stimulated oocytes following first polar body formation (1 h after 1-MA treatment) [Endogenous +1-MA 1 h], and from blastula embryos (12 h after 1-MA treatment) [Endogenous +1-MA 12 h].

After (Line 433):

Sanger sequencing results of the 3’ terminal region of cDNA from endogenous maternal cyclin B mRNA of oocytes at Pro-I without 1-MA treatment [Endogenous −1-MA], from stimulated oocytes following first polar body formation (1 h after 1-MA treatment) [Endogenous +1-MA 1 h], and from blastula embryos (12 h after 1-MA treatment) [Endogenous +1-MA 12 h]. The mean number ± SE of uridine residues in the five nucleotides from the 3′ end was 0.0 ± 0.0 for [Endogenous −1-MA], 2.6 ± 0.5 for [Endogenous +1-MA 1 h], and 1.4 ± 0.3 for [Endogenous +1-MA 12 h]. A Tukey HSD test revealed that the mean number in the [Endogenous −1-MA] group was significantly lower than those in the other two groups (p < 0.05).

In addition, we revised Figure 2 C legend as follows:

Before:

Sequencing results of the 3’ terminal region of cDNA of endogenous maternal Rps29 mRNA from oocytes at Pro-I without 1-MA treatment [Endogenous −1-MA], stimulated oocytes after the first polar body formation (1.5 h after 1-MA treatment) [Endogenous +1-MA 1.5 h], and blastula embryos (12 h after 1-MA treatment) [Endogenous +1-MA 12 h].

After (Line 498):

Sequencing results of the 3’ terminal region of cDNA of endogenous maternal Rps29 mRNA from oocytes at Pro-I without 1-MA treatment [Endogenous −1-MA], stimulated oocytes after the first polar body formation (1.5 h after 1-MA treatment) [Endogenous +1-MA 1.5 h], and blastula embryos (12 h after 1-MA treatment) [Endogenous +1-MA 12 h]. The mean number ± SE of uridine residues in the five nucleotides from the 3′ end was 0.0 ± 0.0 for [Endogenous −1-MA] (poly(A) length >40 residues), 1.9 ± 0.5 for [Endogenous +1-MA 1.5 h] and 0.4 ± 0.2 for [Endogenous +1-MA 12 h] (poly(A) length >40 residues). A Tukey HSD test revealed that the mean number in the [Endogenous −1-MA] group was significantly lower than the [Endogenous +1-MA 1.5 h] group (p < 0.05).

Comments 5:

Discussion: This discussion places the findings well into the existing literature and points toward the role of uridylation other than degradation. This manuscript could benefit from further consideration of the possible molecular mechanisms that account for the differences in stability among uridylated mRNAs. In addition, a short discussion of how these findings might inform future research on mRNA stability in other marine organisms would increase the scope of this study. The limitations section is superficial, with important biases, especially in the selection of particular stages in development.

Responses 5:

Thank you for your valuable comments. In response, we have revised the text to include “marine organisms” and “other stages of development” in the following sentence:

Before:

Further research is required to improve our understanding of the recycling [56] or re-elongation of uridylated mRNAs in various animals.

After (Line 832):

Further research is required to improve our understanding of the recycling [54] and re-elongation of uridylated mRNAs in other developmental stages of starfish embryos, as well as across a broader range of animals, including marine organisms.

Comments 6:

Conclusion: The conclusions were drawn to match the data presented. The authors amply express in the text that, in starfish, the uridylated mRNAs can be reclaimed for translational activity with the addition of a poly tail, thus providing an important insight into the topic at hand. This section can be further upgraded by suggesting studies that would confirm whether such post-transcriptional processes occur in other organisms.

Responses 6:

We added Conclusions section:

In starfish oocytes, uridylated maternal cyclin B mRNAs remain stable and are polyadenylated to enable translation following hormonal stimulation. These mRNAs are deadenylated and re-uridylated, ultimately undergoing decay at the blastula stage. In contrast, maternal ribosomal protein mRNAs, such as Rps29 and Rpl27a, are polyadenylated and translationally active prior to hormonal stimulation. Following stimulation, these mRNAs become deadenylated, uridylated, and translationally inactive. At the morula stage, however, the uridylated maternal ribosomal protein mRNAs are re-polyadenylated, restoring their translational activity. These findings suggest that uridylated mRNAs in starfish adopt a poised state, allowing them to be either recycled for translation or targeted for decay.

Comments 7:

   Major Flaws:

  • The study did not consider the possible role of uridylation in non-maternal mRNAs, which would extend the generality of the findings.

Responses 7:

As suggested, we added possible post-transcriptional processes in other organisms as shown in our answer for your comment in the previous your comment of 4.

Comments 8:

  • It should be better hypothesized how the mechanisms underpin why some uridylated mRNAs are re-polyadenylated.

Responses 8:

Thank you for your insightful comments. While we are not aware of any reports demonstrating the re-polyadenylation of uridylated mRNAs, we proposed a potential mechanism for re-polyadenylation. Specifically, we suggested that the 3′ UTR might play a key role in determining the fate of uridylated mRNAs in Discussion (Line 779): “the 3’ UTR is likely to be involved in the choice of fate and uridylation, as the 3’ UTRs of the ribosomal protein mRNAs and cyclin B mRNA reproduced the pattern of degradation or re-polyadenylation of endogenous mRNAs at the morula, blastula, and gastrula stages. “

Comments 9:

  1. Minor Issues:
  • Explain the rationale behind the choice of time points for hormonal stimulation, such as 1 and 6 h.

Responses 9:

We selected key developmental time points, including GVBD, 1PB formation, and the morula stage, and collected samples for analysis at these specific stages. In particular, we clarified that 1 hour and 6 hours after hormonal stimulation correspond to the 1PB formation and morula stages, respectively, as illustrated in Fig. 1B.

Comments 10:

        Some sentences in the discussion can be simplified in a more direct manner. For example, from lines 411 to 415:.

Responses 10:

As suggested, the sentence (lines 411 to 415) was simplified in a direct manner.

Before: These results suggest that the uridylation of cyclin B mRNA at the blastula stage in starfish embryos may function to induce the degradation of the mRNAs, as was previously demonstrated in vertebrate embryos [4] and that the 3′ UTR of cyclin B is sufficient for the deadenylation and uridylation of the mRNA.

After (Line 414): These results suggest that uridylation of cyclin B mRNA at the blastula stage in starfish embryos promotes its degradation, as seen in vertebrate embryos [4]. They also show that the 3′ UTR of cyclin B alone is enough to drive its deadenylation and uridylation.

Comments 11:

     Some typing errors in Table S1 need to be corrected.

Responses 11:

Thank you for your comment. This (′) may depend on the Mac or Windows. Using my PC, I cannot see the typing errors, but I checked it and replaced ′.

Comments 12:

        The manuscript provides several issues of concern regarding the similarity indices, an overall similarity score of 75% that is above acceptable thresholds, and a single source-matched problem of 38% . A manuscript should generally have a below-30% overall similarity, as stated in the MDPI's guidelines for publication ethics, although the similarity to any single source should not be more than 15%. These scores require revision, involving substantial improvements, before consideration is made to publish the manuscript.

Responses 12:

We checked the iThenticate report and found that the overall similarity score of 75% was primarily due to the preprint server bioRxiv, where part of our work had been uploaded. Furthermore, we had clearly declared that the content of this manuscript was submitted to a preprint server, which is accepted by Biomolecules. When the similarity was analyzed without including the bioRxiv content, the score was significantly lower. Therefore, we believe this should not pose any issue.

Reviewer 3 Report

Comments and Suggestions for Authors

The manuscript entitled ‘Recycling of uridylated mRNAs in starfish embryos’ presents an interesting topic in the area of reproduction with aim to investigate the fate of maternal cyclin B and ribosomal protein mRNAs during starfish development, in order to analyse the consequences of these components into embryo development.

In general, the manuscript is well written. The introduction is interesting and up-to-date. The objective is clear. The material and methods section adequately describes the procedures to be followed. The results are well presented and discussed in relation to the collected literature. Therefore, I consider the manuscript suitable for publication.

Author Response

We would like to thank the reviewer for their constructive comments. 

Round 2

Reviewer 2 Report

Comments and Suggestions for Authors

No further comments!

Author Response

Thank you very much for your review.